# Immune Responses Induced by mRNA Vaccination in Mice, Monkeys and Humans

**DOI:** 10.3390/vaccines9010061

**Published:** 2021-01-18

**Authors:** Alberto Cagigi, Karin Loré

**Affiliations:** Division of Immunology and Allergy, Department of Medicine Solna, Karolinska Institutet, Karolinska University Hospital, 161 64 Solna, Sweden; alberto.cagigi@ki.se

**Keywords:** mRNA vaccine, dendritic cell, protein translation, type I interferon, Th1 polarization, antibody response, SARS-CoV-2

## Abstract

In this concise review, we summarize the concepts behind mRNA vaccination. We discuss the innate and adaptive immune response generated by mRNA vaccines in different animal models and in humans. We give examples of viral infections where mRNA vaccines have shown to induce potent responses and we discuss in more detail the recent SARS-CoV-2 mRNA vaccine trials in humans.

## 1. Introduction

The first successful experimental injections of mRNA that resulted in translated proteins were performed in murine muscle cells in vitro in the early 1990s [1]. This encouraged the development of mRNA and also DNA as a new concept of vaccination [2,3,4]. However, it was not until critical modifications of the mRNA leading to increased stability and translational capacity were introduced that the field of mRNA vaccination started its real expansion [5]. The interest in mRNA vaccination has been growing substantially the last decade, and has during the current SARS-CoV-2 pandemic further accelerated as some of the first vaccine candidates approved for human use are mRNA vaccines (https://www.raps.org/news-and-articles/news-articles/2020/3/covid-19-vaccine-tracker). A competitive advantage of mRNA vaccines that has been particularly emphasized is that the vaccine production and purification processes are similar, despite the encoded antigens. This feature gives prospects of using harmonized protocols for different mRNA vaccines [6]. Production of new mRNA vaccines can thereby be initiated immediately once the genomic sequence of a target antigen has been identified. This can result in reduced costs and faster production of new vaccines against emerging infectious diseases, as exemplified by the mRNA vaccines against SARS-CoV-2. Concerns with reproducibility of virus cultures and protein production in mammalian cells or issues with egg allergies associated with seasonal influenza vaccines produced in eggs are circumvented by mRNA vaccines. mRNA vaccines, as opposed to DNA vaccines, do not need to be delivered into the cell nucleus via electroporation or other devices, since protein translation from mRNA occurs in the cytoplasm and regular needle injection is sufficient. In addition, there is not a need for vaccine adjuvants, since mRNA itself has the inherent ability to induce a strong innate immune response [7,8]. However, the innate immune activation by mRNA vaccines may also result in side effects and limited protein translation followed by weak adaptive immune responses. Fine-tuning the innate and the adaptive immune responses by modifying the mRNA, the mRNA dosage and the formulation strategy is a major area of research. 

Two forms of mRNA vaccines have primarily been studied. The main group consists of (1) the more conventional chemically modified and unmodified sequence-optimized mRNA vaccines and (2) the self-amplifying mRNA vaccines based on virally derived RNA that encodes both the antigen of interest and the proteins enabling RNA vaccine replication. The advantage of self-amplifying mRNA vaccines is that a lower dose of mRNA can often be used, which also results in decreased (Toll-like-receptor) TLR recognition and associated innate immune responses. Yet, the conventional mRNA vaccines have progressed the furthest into clinical practice so far. Apart from the important improvements in optimizing the mRNA in both groups of mRNA vaccines for translatability, stability and reactogenicity over the years, major advancements have also been made regarding potent and well-tolerated delivery technologies, such as lipid-based drug delivery systems [9,10]. This is a critical part in the development of mRNA vaccines. Early studies showed that even if the mRNA is stabilized, a large proportion is simply filtered out via the kidneys and urine after administration due to its small size [11]. Significant improvements in the bioavailability of mRNA through formulation have since been made. The majority of mRNA vaccines are currently packaged in biodegradable ionizable lipid nanoparticles (LNPs) consisting of variants of phospholipids, cholesterol and polyethylene glycol (PEG) containing lipids [10,12]. The ionizable lipid is positively charged to form complexes with the negatively charged mRNA for protection of the mRNA and may also help with cellular uptake and endosomal escape [12]. PEG–lipids significantly increase the bioavailability, i.e., time of mRNA in the circulation, which greatly improves the prospects for therapeutic use, but this can be at the expense of reduced transfection efficiency [13]. Modifying the PEG–lipids or cholesterol contents can also be utilized to alter the particle size and morphology of the LNP and in turn influence the trafficking routes after administration and the efficiency of mRNA delivery to cells [10,12,13,14]. Furthermore, mixing of unsaturated lipids can also improve the uptake by cells [15]. Alternative nanoparticles described are the core-shell structured lipopolyplex (LPP) platform in which mRNA binds to a positively charged protein or polymer to form a dense core structure that is encapsulated in a lipid shell [16].

Here we review the characteristics of reported immune responses induced by different mRNA vaccines in mouse and non-human primate (NHP) models versus humans. Furthermore, we speculate on the strengths and weaknesses of mRNA vaccines compared to conventional vaccine platforms.

## 2. Innate Immune Activation of mRNA Vaccines

Live-attenuated viruses are amongst the most successful vaccines in eliciting high and long-lasting protection, as illustrated by the measles vaccine and the yellow fever vaccine that can induce antibody levels that are maintained above the protective threshold for decades [17,18]. This might be due to the fact that attenuated infection and viral replication most closely mimic the characteristics of the natural pathogen which results in elicitation of a stronger immune response. Local inflammation at the injection site and recruitment of antigen presenting cells (APCs) are essential to promoting adaptive cellular T cell responses for eliminating infected cells and humoral (antibody) responses for neutralizing pathogens. However, live vaccines may not be administered to immune compromised individuals because of safety risks [19]. Instead, killed/inactivated, split virion or protein-based vaccines are more suitable for a broader population, but are often poorly immunogenic and require an adjuvant to induce sufficient responses. The by-far most clinically used adjuvant, alum, primarily generates a Th2 response. Other adjuvants, such as Toll-like-receptor (TLR) agonists can shift this balance via, for example, production of type I interferon (IFN), which promotes Th1 responses mimicking the milieu of a viral infection [20]. The type I IFN responses induced naturally by the presence of mRNA and the downstream Th1-polarized responses induced are also characteristic for mRNA vaccines [8,21]. This may make mRNA vaccines particularly suitable for viral infections. In addition, upon vaccination with mRNA, the antigenic protein is produced by host cells similar to a viral infection, which can lead to some degree of MHC class I presentation, even if the mRNA sequence is designed to produce secreted or membrane-anchored proteins [22] (Figure 1).

### Vaccine Uptake, Translation and Biodistribution after mRNA Vaccination

Many of the fundamental mechanisms by which mRNA vaccines induce strong responses are incompletely understood. Which cell specific activation contributes the most to vaccine efficacy? What activation may inhibit the generation of adaptive immunity or lead to poor tolerability of the vaccine? In essence, all cells express low-density lipoprotein (LDL) receptors which have been indicated to be used in the uptake of LNPs, facilitated by ApoE lipoprotein which is present in serum and tissues [23,24]. This would suggest that most cells can be recipient cells of LNPs given that they have the appropriate endocytic mechanisms to internalize. Efficient targeting of professional APCs by mRNA vaccines may be one mechanism by which vaccine-specific responses are generated. Mouse studies have demonstrated that direct targeting of dendritic cells (DC) by mRNA was necessary for the induction of antigen-specific T cells [25]. However, direct transfection of APCs has also been proposed to not be required as the antigen can also be produced by muscle cells and further taken up by APCs [26]. In addition, cells that have endocytosed mRNA in LNPs have been shown to secrete extracellular vesicles containing the mRNA which may be an alternative mechanism for delivering mRNA between cells and result in protein translation [27]. 

Using in vivo imaging techniques like the luciferase system applied on mouse models, a number of studies have demonstrated where mRNA vaccines distribute after different routes of administration [23,28,29,30,31]. Intramuscular injection of mRNA vaccines was shown to lead to infiltration of cells to the site of injection and production of the encoded antigen [28]. Detectable protein production was found for up to ten days at the site of injection after subcutaneous, intramuscular and intradermal methods of administration [29]. In contrast, intravenous and intraperitoneal administration resulted in that mRNA translation occurred mainly in the liver and for a shorter duration (up to three days). The very restricted biodistribution to the injection site and its draining lymph nodes with no or limited spread to the kidney, liver, spleen, colon or lung after subcutaneous mRNA vaccination, was also shown by another detection technique based on tomato red expression [32]. The biodistribution of self-amplifying RNA vaccines has been reported to be similar to conventional mRNA vaccines, but showed delayed kinetics and stronger expression of the encoded antigen [33,34].

Using positron emission tomography/computed tomography (PET/CT) and near-infrared imaging, the biodistribution of an mRNA vaccine labelled with a radionuclide/near-infrared probe was assessed in NHP cynomolgus macaques [35]. It was again demonstrated that the vaccine exclusively targeted the injection site and its draining lymph nodes. The vaccine was administered by intramuscular injection in the quadricep muscle of the leg and consequently the vaccine drained specifically into the inguinal, iliac and paraaortic lymph nodes [36]. Similarly, by administration of rhesus macaques with an mRNA vaccine encoding for the fluorochrome mCitrine, the vast majority of the signal was found at the injection site and in its draining lymph nodes [8]. Although neutrophils were found to be the most efficient cells to internalize the mRNA vaccine (delivered in labelled LNPs), monocytes and DCs showed much more efficient translation of mCitrine. As discussed above, whether the mRNA-encoded protein is loaded onto MHC and presented on the cell surface to T cells remains to be elucidated but the APCs that translate protein appear to also specifically undergo maturation characterized by upregulation of co-stimulatory molecules required in the antigen presentation process [8]. The phenotypic maturation may be triggered by the intracellular receptors sensing the uptake of mRNA. Type I IFNs have been reported to support upregulation of molecules involved in antigen presentation and processing in vitro [37,38]. It was shown in vivo that cytokines associated with innate immune activation were primarily induced at the injection site and in the draining lymph nodes after intramuscular injection of mRNA vaccines which indicates a strong local cell activation [39]. Priming of vaccine-specific T cells has also been demonstrated to occur specifically in the vaccine-draining lymph nodes similar to where antigen and cytokine production are detected [8,40] (Figure 2).

As expected, the mRNA content of the vaccine exclusively induced type I IFN, whereas LNPs without mRNA did not have this stimulatory effect [8]. mRNA is recognized by pattern recognition receptors (PRRs) such as TLR3, TLR7, and TLR8 (Figure 1). Double-stranded RNA (dsRNA) can also be sensed by receptors in the cytoplasm such as Retinoic-acid-inducible gene I (RIG-I) and melanoma differentiation-associated 5 (MDA5), which stimulate type I IFN secretion [41]. Excessive immune responses and secretion of large amounts of type I IFN can inhibit the translation of mRNA and thereby reduce the responses to the vaccine [5,42]. On the other hand, type I IFNs have been shown to be critical for inducing anti-tumor responses both in mice and humans in response to intravenously administered RNA aimed at cancer immunotherapy [25]. Other studies utilizing mice devoid of the IFN receptor showed that IFN reduces the antigen expression and consequently the induction of antigen-specific immunity [43]. The contradictory results may be related to differences in the amounts of type I IFN stimulated by the different mRNA constructs and the formulation and route of administration used. The timing of transfection and initiation of mRNA vs. induction of IFN secretion may also be critical for whether there is any reduction of antigen expression [44].

## 3. Antibody Responses by mRNA Vaccines to Infectious Diseases

The correlate of protection of most vaccines is the production of antibodies which therefore represents a central part in the evaluation of immunogenicity of new vaccines. For some infectious diseases, established antibody levels accepted by the WHO exist which can serve as an indicator of protective responses. This offers the possibility to assess the performance of new mRNA vaccine candidates and benchmark against licensed vaccines if available. A major advantage with mRNA vaccines is that the produced protein (given the sequence encodes for the native form of the pathogenic antigen) will have the accurate and same conformation and glycosylation as the live pathogen and thus favoring the development of correct antibody specificities [45]. Antibody levels and quality upon vaccination with LNP-formulated mRNA may also be influenced by the size of the LNPs (range up to 200 nm) that enables both cellular and passive transport to the draining lymph nodes which means that B cells at these locations can potentially be directly targeted by the vaccine [46]. B cells are able to take up LNPs and produce protein antigen from the mRNA, but this function appears to be less pronounced compared to monocytes and DCs [8]. Instead, the B cells may predominantly interact via their cognate B cell receptors with antigens expressed or secreted by adjacent cells. The architecture of lymph nodes favors efficient interactions of antigens and cognate B cell receptors leading to the cascade on immunological events including selection and differentiation of B cells into antibody producing cells [47,48]. B cells located in mRNA vaccine-draining lymph nodes have been shown to exhibit a more activated phenotype and germinal center formation and plasmablast formation have been demonstrated at this site [21,31,49] (Figure 2). Similarly, T follicular helper and germinal center B cell responses after mRNA vaccination of mice were shown to significantly expand and to directly correlate with the magnitude and durability of antibody responses [49]. A simplified overview of the events that follow uptake of an LNP-formulated mRNA vaccine and generation of adaptive immune responses is presented in Figure 2. A considerable benefit with mRNA vaccines may be that protein production occurring over an extended period would allow for prolonged antigen: B cell interaction which can refine the B cell response in terms of selection of better affinity B cell clones and driving somatic hypermutations [50], although this remains to be proven after mRNA vaccination. 

Below we will discuss some major mRNA vaccine candidates (grouped by different viral infections) that have progressed towards clinical testing. Important to note is that there are indeed reports of several other promising mRNA vaccine candidates tested in preclinical models that are not brought up in this concise review.

### 3.1. Rabies

One of the early-generation mRNA vaccines that has been extensively studied was developed by one of the largest mRNA vaccine companies, CureVac AG (Tübingen, Germany), and was based on sequence-optimized, chemically unmodified mRNA encoding the rabies virus glycoprotein (RABV-G). This mRNA vaccine was initially used in the vaccine formulation RNActive^®^ with protamine for enhanced adjuvanticity and was shown to be highly immunogenic [51]. Intradermal administration in mice resulted in antibody levels above the level suggested by the WHO to be required for protection and lasted above this threshold for over one year which were higher than the levels induced by the clinically used rabies vaccine Rabipur. This mRNA vaccine also conferred full protection in a murine challenge model of intracerebral inoculation with rabies virus. In addition, adult and newborn pigs induced protective neutralizing antibody levels after mRNA vaccination [51]. The thermostability of the vaccine was further shown to be high since multiple cycles of cold chain interruption did not affect the immunogenicity or its efficacy to protect mice against lethal challenge with rabies virus [52]. This vaccine became the first mRNA vaccine to be tested in humans. The clinical study evaluated doses ranging from 80 to 640 μg, using either needle-free and needle injections by the intramuscular or intradermal route with immunizations at day 0, 7 and 28 [53]. The majority of the individuals receiving 80 μg showed responses by day 21 but some as early as day 7. The responses in most of the individuals reached the WHO threshold for protection. However, in contrast to mice the antibody levels were undetectable in most of the participants after a year from vaccination [53]. Although there were mild-to-moderate reactions at the injection site reported and some systemic adverse events including fatigue and fever within a week after vaccination the vaccine was generally well tolerated and the study is a milestone in the development of mRNA vaccines as a concept [53]. This vaccine was shown to be improved in immunogenicity and durability by changing to a LNP formulation and using the intramuscular delivery route [39]. Cynomolgus monkeys immunized with the updated mRNA/LNP vaccine formulation given at 10 μg and 100 μg induced higher antibody titers compared to the full human dose of Rabipur. The neutralizing titers against rabies virus were maintained above the WHO level for 182 days. This vaccine is currently in phase I clinical testing in humans (ClinicalTrials.gov: NCT03713086).

### 3.2. Influenza

Vaccination against influenza, primarily pandemic strains, has been a major area studied in the mRNA vaccine field. Since the culture of new pandemic influenza viruses in vitro for use as inactivated or split virus vaccines can take several months, considerable time can be gained by using mRNA vaccines that can be produced quickly after the release of a new viral sequence [54]. This represents a typical example of when the mRNA vaccine platform can have a competitive advantage over other vaccine platforms. The studies of mRNA vaccines against influenza are also facilitated by the availability of suitable animal models, benchmark comparison to licensed vaccines and accepted antibody levels for correlate of protection of seasonal influenza. Influenza vaccines based on sequence-optimized, chemically unmodified mRNA, modified mRNA and self-amplifying RNA have all shown promising results.

The first study showing the immunogenicity and protection induced by an unmodified mRNA vaccine encoding various influenza antigens, including hemagglutinin (HA) of 3 different influenza viruses and the conserved nucleoprotein, was performed in mice, ferrets and pigs where T cell and B cell responses could be directly compared to those elicited by licensed inactivated virus vaccines [55]. Furthermore, a similar mRNA vaccine encoding influenza H1N1 hemagglutinin formulated in LNPs demonstrated to be highly immunogenic and induced strong antibody responses in cynomolgus monkeys [39]. Doses given at 10 μg and 100 μg induced higher antibody titers compared to the full human dose of the clinically licensed vaccine Fluad. The H1N1- hemagglutinin inhibition (HAI) titers were maintained above the threshold considered as a correlate of protection for seasonal influenza for a year after 10 μg of mRNA vaccine encoding for H1N1.

An LNP formulated modified mRNA vaccine against H10N8 and H7N9 influenza viruses developed by Moderna Inc (Cambridge, MA, USA), another leading mRNA vaccine company, also induced high titers of antibodies above the reported threshold for protection and seroconversion as measured by both HAI and microneutralization (MN) assays in mice, ferrets, and NHPs [30]. The antibody levels were able to protect mice against lethal influenza challenge and reduced lung viral load in ferrets. A deeper characterization of the immune responses to the mRNA vaccine encoding H10N8 was also performed showing induction of HAI titers above the threshold of correlate of protection using 50 μg [8]. Intradermal administration generated slightly higher antibody levels directly after immunization compared to intramuscular administration. Germinal centers were formed in the vaccine-draining lymph nodes along with an increase in circulating T follicular helper cells (cTfh) with a Th1 phenotype [21]. Plasmablasts appeared transiently after the boost immunization and H10-specific plasma cells established rapidly in the bone marrow after the first immunization and persisted over time. H10-specific memory B cells were also generated fast but waned over time. A human dose-escalating phase 1 study (25–400 μg) was performed with this vaccine which showed a safe and well-tolerated profile and induced antibody titers above the reported threshold for protection in the high-dose groups and detectable titers 6 months post-vaccination [30,56].

When BioNTech Pharmaceuticals (Mainz, Germany), another large mRNA company, compared synthetic mRNA and self-amplifying -RNA expressing influenza hemagglutinin of three different strains (H1N1, H3N2 and B), both vaccines induced protective antibody titers in mice, but much lower (64-fold) dose of self-amplifying-RNA was needed to induce comparable protection [33].

### 3.3. Zika

During the 2015 Zika virus epidemic in Brazil, multiple mRNA vaccines against Zika virus were developed. Two different mRNA vaccines encoding the pre-membrane and envelope glycoproteins of Zika virus showed high levels of neutralizing antibodies and protection after lethal challenge in mice and in rhesus macaques [57,58]. In one of these studies, blocking type I IFN was also evaluated which showed no differences in the vaccine response [57]. After antibody-dependent enhancement of the pathology after Zika virus exposure and cross-reactivity with Dengue virus infection were found, a new mRNA vaccine encoding a modified Zika pre-membrane and envelope antigens, was designed. This vaccine remained to induce protective responses against Zika challenge but was also able to rescue fetal viability after maternal infection in mice [59]. This mRNA vaccine is currently being tested in a clinical trial (ClinicalTrials.gov: NCT03014089). A self-amplifying mRNA vaccine encoding the same antigens, formulated in LNP, was also tested in mice and resulted in protective and long-lasting immune responses [60].

### 3.4. CMV, HIV-1, RSV and Ebola

Different combinations of modified mRNA/LNP vaccines encoding the CMV glycoproteins gB and the pentameric complex (PC) elicited potent neutralizing antibody titers in mice and cynomolgus macaques and the titers were well maintained after 1, 2 or 3 doses during the study period of 222 days. The levels of neutralizing antibodies measured in mice and macaques were benchmarked against the levels reached as a result of administration of a human dose of Cytogam, the clinically used formulation of IgG for CMV prophylaxis, and against the levels measured in sera of CMV+ human donors. Higher or similar antibody titers and neutralization were observed after vaccination as compared with these control groups [61]. Another CMV vaccine based on self-replicating RNA (expressing hCMV pp65-IE1 formulated with cationic nanoemulsion (CNE) also induced both neutralizing antibodies and CD4+ T cell responses in rhesus macaques [28]. An alternative approach for prevention of CMV disease in humans with the mRNA technology that has been tested in humans is administration of autologous DCs electroporated with mRNA encoding CMV pp65 ex vivo which resulted in induction or expansion of pp65-specific T cell responses [62]. This approach has also been tested in multiple clinical studies on HIV-1-infected individuals by giving autologous DCs electroporated with mRNA encoding various HIV-1 antigens for enhancement of HIV-1 specific T cell immunity [63,64,65,66].

Chemically modified mRNA vaccines have been tested to express different forms of the RSV F protein, including secreted, membrane associated, prefusion stabilized, and non-stabilized structures, for comparison of antigen conformations in rodent models. Vaccination with mRNA encoding native RSV F elicited antibody responses to both prefusion- and postfusion-specific epitopes, suggesting that this antigen may adopt both conformations in vivo. Incorporating prefusion stabilizing mutations further shifted the immune response toward prefusion-specific epitopes, but did not impact neutralizing antibody titer [45].

Two vaccines encoding either the full-length Ebola virus glycoprotein (GP) or a modified version with the human IgK signal peptide elicited neutralizing antibodies in Guinea Pigs resulting in 100% protection after a lethal guinea pig-adapted Ebola virus challenge. In this model, high levels of the modified vaccine-induced neutralizing antibodies against Ebola virus were detected after 42 days of vaccination [67].

### 3.5. SARS-CoV-2

The first data on safety, tolerability and immunogenicity after SARS-CoV-2 mRNA vaccination in humans were reported by BioNTech [68]. The study subjects received two immunizations of either 10 μg, 30 μg or 100 μg of nucleoside-modified mRNA vaccine encoding a secreted trimerized version of the receptor-binding domain (RBD) of the spike glycoprotein of SARS-CoV-2. RBD-binding antibody titers and neutralizing titers of SARS-CoV-2 were dose-dependent and increased with the boost immunization. The neutralizing titers were in the level of or higher than the antibody levels in convalescent sera from humans who had recovered from COVID-19. However, it is important to note that the level of antibodies needed for protection against SARS-CoV-2 is not yet known. They subsequently found that a similar formulation but encoding the membrane-anchored spike glycoprotein stabilized in the prefusion conformation induced equally high responses but showed less side effects and this vaccine was therefore pursued over the RBD-encoding vaccine to progress further in clinical testing [69]. The antigen-binding titers and neutralizing titers were lower in 65–85 years old compared to 18–55 years old found by both the vaccines. This was the first mRNA vaccine to be approved for human use both by FDA and the European Medicines Agency (EMA).

The enhanced expression and increased immunogenicity gained by using the stabilized prefusion conformation of the spike glycoprotein were also utilized by Moderna Inc in their mRNA vaccine platform against SARS-CoV-2. Early mouse studies showed that this vaccine induced potent neutralizing antibody responses to both wild-type and mutant SARS-CoV-2 strains and protected against SARS-CoV-2 infection in the respiratory tract [70]. This was followed by evaluation of the immunogenicity and protection in rhesus macaques receiving 10 or 100 μg of the vaccine [71]. Again, the responses were dose-dependent and increased with the second immunization. The neutralizing titers exceeded that of convalescent sera from humans who had experienced COVID-19. In addition, there was protection from viral infection in the upper and lower airways, and no pathology detected in the lungs of the rhesus macaques. A phase I trial was conducted using 25, 100 and 250 μg of mRNA vaccine. After the first immunization, neutralizing activity was detected in less than half of the study subjects, but all showed increased antibody titers and neutralization after the second immunization and there was no difference between the two highest vaccine doses [72]. The vaccine at 10 or 100 μg was also tested in an older population (more than 71 years old) which showed dose-dependent responses and that two immunizations were required to reach neutralizing activity and to reach similar levels of antibodies as observed in convalescent sera from COVID-19 patients. The responses in the elderly receiving 100 μg were similar as in individuals between 18–55 years of age receiving this dose [73]. However, the number of study subjects were limited to make solid conclusions on this matter.

CureVac’s SARS-CoV-2 mRNA vaccine, although based on unmodified mRNA, is also encoding pre-fusion stabilized spike protein presented on the cell surface. This vaccine was also found to be highly immunogenic in mice and hamsters and induced strong neutralizing titers and full protection [74]. Two immunizations of 8 μg of this vaccine was required to induce well-detectable spike and RBD-binding antibodies as well as neutralization which resulted in protection from enhanced disease after challenge [75]. This vaccine was tested in a clinical phase I trial at 2, 4, 6, 8 and 12 μg. As observed with the other mRNA vaccines against SARS-CoV-2 there was a clear increase in titers with the boost immunization. The highest dose showed comparable titers to COVID-19 patients [76] and was selected for further evaluation in phase II/III studies. Noteworthy, several additional mRNA vaccines are being tested at different levels of advanced stages (https://clinicaltrials.gov/). Different versions of self-replicating mRNA vaccines are developed and are tested in early clinical trials by Imperial College London/Morningside Ventures (London, UK) and Duke-NUS Medical School/Arcturus Therapeutics (Singapore) and via a collaboration between University of Washington, National Institutes of Health Rocky Mountain Laboratories, HDT Bio Corp (Seattle, WA, USA), and Gennova Biopharmaceuticals (Pune, India). Other mRNA vaccines are developed by Chulalongkorn University and Faculty of Tropical Medicine, Mahidol University (Nakhon Pathom, Thailand); Academy of Military Medical Sciences, Suzhou Abogen Biosciences Yunnan Walvax Biotechnology Co., Ltd. (Yunnan, China); Stemirna Therapeutics (Shanghai, China) and Zydus Cadila (Ahmedabad, India) and several more efforts take place around the world. There are also multiple studies planned to test the approved mRNA vaccines in specific age groups and patient groups with different immunological conditions.

An important general finding for the field is that mice, NHPs and humans immunized with different mRNA vaccines encoding pre-fusion stabilized spike protein not only developed neutralizing antibodies against the original SARS-CoV-2 strain but also the D614G mutant strain [77]. Humans immunized with the approved mRNA vaccine from BioNTech were found to induce neutralizing antibodies against the N501Y mutant strain [78].

## 4. T Cell Responses by mRNA Vaccines

T cell immunity is usually not a correlate of protection after vaccination but CD4+ T cells are required to support B cell differentiation and establish memory responses. Since mRNA vaccines to some extent mimics viral infection they could potentially also promote CD8+ T cell responses. A number of studies have evaluated the T cell responses induced by mRNA vaccines.

The sequence-optimized, chemically unmodified mRNA vaccine encoding for RABV-G mentioned above induced detectable RABV-G-specific CD4+ and CD8+ T cells, evidenced by IFN-γ, TNF, IL-2 and CD107a expression after peptide stimulation in vitro, in mice and pigs. The CD4+ T cell response was found to be higher than the responses induced by the licensed vaccine Rabipur [51]. The same vaccine subsequently tested in the first human trial induced RABV-G -specific CD4+ T cells transiently detected after 3 doses of the vaccine [53].

Similarly, mRNA vaccines encoding for either RABV-G or influenza H1N1 hemagglutinin but formulated in LNPs also induced well-detectable CD4+ T cell responses in mice and in cynomolgus monkeys [39]. However, in terms of CD8+ T cells, it is clear that such responses are much more apparent in mice, whereas they often are not detectable in monkeys or humans.

T cell responses were detected in mice after vaccination with an LNP formulated modified mRNA vaccine encoding H10N8 and H7N9 influenza hemagglutinin [30]. Detectable CD4+ T cell- and not CD8+ T cell- responses were induced in rhesus macaques to this vaccine [8,21], but the clinical phase I study did not find detectable T cell responses in vaccinated individuals [30,56]. The fact that HA is not a robust T cell antigen was speculated as a reason for the lack of detectable T cell responses.

A modified mRNA/LNP vaccine that encoded the CMV glycoproteins gB and the pentameric complex (PC) and included the immunodominant T cell antigen pp65 was also shown to elicit robust T cell responses in mice, as verified by overlapping peptide stimulation and intracellular IFN-γ, TNF and IL-2 [61].

Robust T cell responses to SARS-CoV-2 infection have been readily detected [79]. However, whether T cell-mediated protection via vaccination can occur and be of importance is still unknown [80]. The elicitation of T cell responses by mRNA vaccines against SARS-CoV-2 has been assessed in mice, in NHPs and in humans. The mRNA vaccines described above from CureVac, Moderna and BioNTech all elicited strong CD4+ and CD8+ T cells with Th1-type biased responses in mice [70,74,81]. Robust Th1-type CD4+ and CD8+ T cells in mice after administration of one or two doses of other LNP-formulated nucleoside-modified mRNA vaccines encoding the full-length spike protein or the RBD were also detected [82,83]. RBD-specific IFN-γ responses were also detected in cynomolgus macaques in response to another mRNA vaccine after two immunizations [83]. Spike-specific total T cell responses were detectable in rhesus macaques by IFN-γ Elispot 13 days after the first immunization with CureVac’s vaccine. The responses waned but increased after the second dose [75]. Rhesus macaques vaccinated with Moderna’s mRNA vaccine induced a Th1-biased CD4+ T cell response detected by intracellular cytokine staining, but low or undetectable Th2 or CD8+ T cell responses were found [71]. Humans immunized with Moderna’s vaccine showed similar responses as in rhesus macaques [73]. In that study, participants of two age groups were vaccinated (between 56 and 70 years old and above 71 years old) with either 100 or 25 μg of mRNA. Th2 and CD8+ T cell responses were minimal independently of age and were measurable only after two vaccinations with 100 μg [73].

## 5. Conclusions

Since the idea of injecting mRNA with the purpose of vaccination was first tested in the early 1990s [1], a much more intimate understanding of the immunological events from administration to generation of a response has been acquired, as summarized in this concise review. Despite the several advantages of mRNA vaccines that have become apparent over the years, the approval of the first mRNA vaccine for human use was not until the SARS-Cov-2 pandemic significantly accelerated and expedited clinical testing and review. This undoubtedly represents a milestone in the history of vaccination, and if successful, mRNA-based vaccines may serve as the prompt “standard” solution for future pandemics but may also replace some regular conventional protein-based and live-attenuated vaccines. The mRNA platform may also be superior over other platforms to be most rapidly modified and distributed to combat new mutated virus strains appearing during a pandemic. An mRNA vaccine may potentially also contain a mix of multiple sequences for broad coverage.

On the other hand, data still need to be generated to assess whether mRNA vaccination is suitable for all people, including children, elderly, and immunosuppressed individuals, and patients with chronic conditions such as autoimmune disorders. The compatibility with different medical drugs also needs to be assessed. Will the type I IFN responses induced by mRNA vaccines be an issue for people with various underlying conditions or for people who are on type I IFN treatment? Some of these types of investigations have already been initiated [73] or are planned. Other issues with mRNA vaccines that have caused significant concerns in the health care system during the distribution of the approved SARS-CoV-2 mRNA vaccines are the cold chain and storage restrictions. Many outpatient clinics and vaccination sites, even in high income countries, do not have access to low temperature freezers to meet the requirements of some these mRNA vaccines. This challenge would be even more pronounced in low-income countries. There is therefore an urgent need for improvements and validation on this matter.

In addition to using mRNA as a prophylactic vaccine for inducing antibodies and protecting against infectious diseases, other applications of mRNA might also be further developed—which was not the topic of this review—such as mRNA vaccination for the treatment of cancer or administration of mRNA for directly encoding human neutralizing monoclonal antibodies as an alternative to passive vaccination/monoclonal antibody treatment for ongoing infections. This has already been tested against Chikungunya virus (CHIKV) where mRNA encoding for the monoclonal antibody CHKV-24 was shown to be successful in protecting mice and cynomolgus macaques from developing disease [84].

The future for mRNA vaccines appears bright, and the knowledge on detailed immunological mechanisms by which they work will likely expand substantially over the coming years. 

## Figures and Tables

**Figure 1 vaccines-09-00061-f001:**
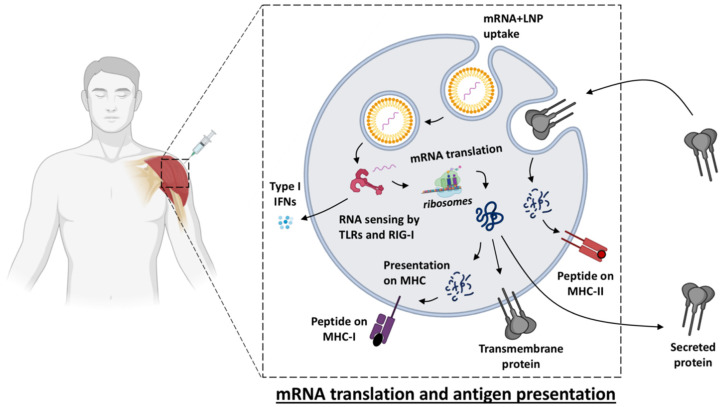
Simplified overview of the events that follow uptake of an LNP-formulated mRNA vaccine by a cell. Once the mRNA molecule is released from the LNP into the cytosol, it is sensed by toll-like receptors (TLR), e.g., TLR3 or 7/8 and by retinoic acid-inducible gene (RIG)-I, which promotes secretion of type I interferons (IFNs) to the extracellular matrix that will create a milieu that favors Th1 responses over Th2. mRNA is directly translated by ribosomes into polypeptides which are processed by the proteasome system, leading to peptide presentation onto MHC-I on the cell surface (similarly as during a viral infection), and post-translationally modified to be folded into the protein which, depending on the mRNA design, can either be membrane-anchored or be secreted. Peptide presentation onto MHC-II may occur on APCs after protein uptake of extracellular proteins or of cell debris containing protein. This figure was created using BioRender.com.

**Figure 2 vaccines-09-00061-f002:**
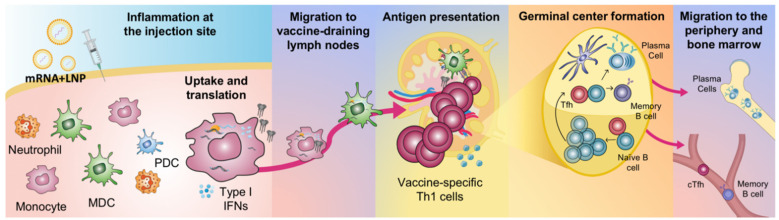
Proposed sequence of events leading to the generation of adaptive immune responses upon mRNA vaccination. Local inflammation at the injection site promotes the infiltration of immune cells, including neutrophils, monocytes, myeloid dendritic cells (MDCs) and plasmacytoid dendritic cells (PDCs). Neutrophils can efficiently take up LNPs, but monocytes and MDCs translate mRNA more efficiently. Secretion of type I interferons (IFNs) is stimulated. mRNA/LNP and protein antigen will disseminate and cells will migrate to the vaccine-draining lymph nodes. Antigen presentation to T cells and interactions of antigen and B cells take place at these sites. This leads to the formation of germinal centers, which results in the generation of memory B cells and antibody-producing plasma cells that reside to the bone marrow. The basic illustration was made by Elizabeth Thompson and some elements were created using BioRender.com.

## Data Availability

Not applicable.

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
