# Peer review of "Immune Responses Induced by mRNA Vaccination in Mice, Monkeys and Humans"

_vaccines, 2021, doi:10.3390/vaccines9010061_

Round 1
Reviewer 1 Report
The authors have written a very comprehensive review summarizing several essential aspects of mRNA vaccines, discussing them in the context of various infectious diseases, including their mechanism of action, it’s potential advantages and limitations as compared to conventional protein-based vaccines. With minimum jargon use, the authors have explained how mRNA vaccines trigger the innate and adaptive immune response. The review is effectively categorized and introduces the researcher to the field of mRNA vaccines using impressive graphics.
The only minor comment I have is that it is recommended to add a short discussion on why lipid nanoparticles or lipoplexes are used to deliver the mRNA vaccines and how is PEGylation or polymer coating helping in delivery. A few lines towards the end of the second last paragraph of the introduction is recommended. Here are some suggestions for references along this line: https://pubs.acs.org/doi/10.1021/acs.nanolett.0c01386
https://www.future-science.com/doi/10.4155/tde-2016-0006
https://www.liebertpub.com/doi/full/10.1089/nat.2018.0721
https://onlinelibrary.wiley.com/doi/abs/10.1002/anie.202013927?casa_token=EbBM0al4ITsAAAAA:gAIVM3DSJ8QvZLK7aUzCA6fTAQ0XBOEyTGIXY9lWnQU-KV2LuEshviWsaQ4tZhyuP7fnGFzSaRiFOP4
Author Response
Reviewer 1
The authors have written a very comprehensive review summarizing several essential aspects of mRNA vaccines, discussing them in the context of various infectious diseases, including their mechanism of action, it’s potential advantages and limitations as compared to conventional protein-based vaccines. With minimum jargon use, the authors have explained how mRNA vaccines trigger the innate and adaptive immune response. The review is effectively categorized and introduces the researcher to the field of mRNA vaccines using impressive graphics.
The only minor comment I have is that it is recommended to add a short discussion on why lipid nanoparticles or lipoplexes are used to deliver the mRNA vaccines and how is PEGylation or polymer coating helping in delivery. A few lines towards the end of the second last paragraph of the introduction is recommended. Here are some suggestions for references along this line: https://pubs.acs.org/doi/10.1021/acs.nanolett.0c01386, https://www.future-science.com/doi/10.4155/tde-2016-0006https://www.liebertpub.com/doi/full/10.1089/nat.2018.0721https://onlinelibrary.wiley.com/doi/abs/10.1002/anie.202013927?casa_token=EbBM0al4ITsAAAAA:gAIVM3DSJ8QvZLK7aUzCA6fTAQ0XBOEyTGIXY9lWnQU-KV2LuEshviWsaQ4tZhyuP7fnGFzSaRiFOP4
A: Thank you for the constructive criticism on our review article. We have revised the manuscript accordingly. We thank you for the insightful comments which helped us to improve the text. We are grateful you pointed us to these articles and we have now reviewed them and some more and added several sentences on this topic in the second last paragraph of the introduction as suggested.
Reviewer 2 Report
In the paper the authors summarize how RNA vaccines work and give several successful examples used in different animal models and humans.The paper is very clearly written and the illustrations are really clear and helpful. I just have some minor suggestions:
It is very well described in the text the advantages of mRNA vaccines but not much is mentioned about the disadvantages. Maybe something could be added in that regard.
In line 221, could you add the reference for those experiments?It is not clear from the text if it is the same vaccine used in reference 44, otherwise it could be rephrased
In line 235, could you determine which other routes are used for the delivery?
Are there other mRNA vaccines? if yes, why the ones described were chosen for the review and no others?
Could the SARS2 section be completed with more vaccines maybe?
I would really suggest to do a table summarizing which vaccine is used, animal model used, efficacy, how long does it last, etc.
Author Response
Reviewer 2
In the paper the authors summarize how RNA vaccines work and give several successful examples used in different animal models and humans. The paper is very clearly written and the illustrations are really clear and helpful. I just have some minor suggestions:
It is very well described in the text the advantages of mRNA vaccines but not much is mentioned about the disadvantages. Maybe something could be added in that regard.
A: Thank you for the constructive criticism on our review article. We have revised the manuscript accordingly. We thank you for the insightful comments which helped us to improve the text. We have now commented on the major disadvantage of the potential cold chain and storage issue in the conclusion remarks.
In line 221, could you add the reference for those experiments? It is not clear from the text if it is the same vaccine used in reference 44, otherwise it could be rephrased
A: Thank you for pointing this out. We have now clarified this.
In line 235, could you determine which other routes are used for the delivery?
A: Thank you for pointing this out. We have now clarified this.
Are there other mRNA vaccines? if yes, why the ones described were chosen for the review and no others?
A: We have now expanded the section on SARS-CoV-2 mRNA vaccines and mentioned other candidates in late stage development by others than Moderna, BioNTech and CureVac. It is also highlighted in the review that we focus on mRNA vaccines that have reached the clinic or clinical trials since there are a high volume of preclinical candidates which would be impossible to go through in detail in the review.
Could the SARS2 section be completed with more vaccines maybe?
A: Please see above.
I would really suggest to do a table summarizing which vaccine is used, animal model used, efficacy, how long does it last, etc.
A: While we certainly agree that a table covering all vaccines, animal models and results would be great to have, we are reluctant to embark on putting such together since there is an overwhelming number of animal studies and we would risk miss information and not cite people’s work fairly. In addition, many studies use their own read-out methodology and therefore direct comparisons are also hard to make.
Reviewer 3 Report
The topic is very interesting, and the review is well written, organized and provides valuable knowledge for the readers. I do not see any major limitation in the draft and would endorse the publication of this manuscript.
Minor comments:
(1). In the introduction, while authors introduce the concept of mRNA vaccine and RNA delivery vectors e.g., lipid nanoparticles, perhaps it would be worth mentioning the delivery of exogenous mRNA and translation into a new protein. For example, delivery of mRNA via lipid nanoparticles (LNPs), uptake of LNP-mRNA, endosomal escape, and translation into protein (PMID: 31551417).
(2). On a separate note, in concluding remarks and future directions, perhaps it would be worth mentioning/highlighting the challenges that mRNA will face against rapidity mutating viruses, and what could be possible improvements in such mRNA vaccines that would be still affective despite of viral mutations.
Author Response
Reviewer 3
The topic is very interesting, and the review is well written, organized and provides valuable knowledge for the readers. I do not see any major limitation in the draft and would endorse the publication of this manuscript.
Minor comments:
(1). In the introduction, while authors introduce the concept of mRNA vaccine and RNA delivery vectors e.g., lipid nanoparticles, perhaps it would be worth mentioning the delivery of exogenous mRNA and translation into a new protein. For example, delivery of mRNA via lipid nanoparticles (LNPs), uptake of LNP-mRNA, endosomal escape, and translation into protein (PMID: 31551417).
A: Thank you for the constructive criticism on our review article. We have revised the manuscript accordingly. We thank you for the insightful comments which helped us to improve the text. We are grateful you pointed us to this article and we have now commented this possible alternative mechanism for delivery of mRNA between cells and cited the publication.
(2). On a separate note, in concluding remarks and future directions, perhaps it would be worth mentioning/highlighting the challenges that mRNA will face against rapidity mutating viruses, and what could be possible improvements in such mRNA vaccines that would be still affective despite of viral mutations.
A: We have now expanded the concluding remarks with a comment on this. We are also citing two articles on neutralizing antibodies induced by SARS-CoV-2 mRNA vaccines that cover mutant strains.